# Topologically distinct 2D and 3D intratumoral heterogeneity scores for preoperatively predicting invasiveness in stage I lung adenocarcinoma: A multicenter study

Zhichao Zuo[1], Xiaohong Fan[2], Ying Zeng[1], Wanyin Qi[3], Wen Liu[4], Wei Li[4]*, Qi Liang[4]*

**1** Department of Radiology, Xiangtan Central Hospital, Xiangtan, Hunan, China, **2** College of Mathematical Medicine, Zhejiang Normal University, Jinhua, Zhejiang, China, **3** Department of Radiology, the Affiliated Hospital of Southwest Medical University, Luzhou, Sichuan, China, **4** Department of Radiology, The Third Xiangya Hospital of Central South University, Changsha, Hunan, China

☯ These authors contributed equally to this work.

\* weililx@csu.edu.cn (WL); csuliangqi10@163.com (QL)

## Abstract

This multicenter study aims to enhance the preoperative prediction of pathological invasiveness in clinical stage I lung adenocarcinoma (LUAD) by developing and validating topologically distinct 2D and 3D intratumoral heterogeneity (ITH) scores derived from chest CT imaging. Patients with histopathologically confirmed LUAD were enrolled from three medical centers. We established a dual-scale computational framework to quantify ITH: the 2D ITH score was derived by integrating local radiomics features with global pixel distribution patterns on the largest cross-sectional slice, while the 3D ITH score captured volumetric heterogeneity using a voxel-based topology-aware approach. Subsequently, six machine learning models integrating clinicoradiologic (CR) features with these heterogeneity scores were developed. Model performance was optimized based on the area under the curve (AUC) across a training set and validated in both an internal test set and an independent external validation set. A total of 1,238 eligible patients were enrolled. Centers 1 and 2 provided 1,053 patients (Training: n=737; Internal Test: n=316), while Center 3 provided 185 patients for external validation. The CatBoost classifier integrating 2D/3D ITH scores with CR features (2DITH-3DITH-CR CatBoost) exhibited superior diagnostic performance, achieving AUCs of 0.867 in the internal test set and 0.881 in the external validation set. The integration of topologically distinct 3D ITH scores significantly improves the preoperative stratification of LUAD invasiveness. The 2DITH-3DITH-CR CatBoost model serves as a robust, non-invasive tool to guide individualized surgical decision-making in clinical practice.

**Data availability statement:** All relevant data included in this study are available and can be accessed by requesting a detailed research proposal to the Ethics Committee/Data Access Committee at the Third Xiangya Hospital of Central South University (dbxy3yy@163.com).

**Funding:** This work was supported by the National Science Fund for Distinguished Young Scholars (Grant No. 2023JJ10091 to WL). The funders had no role in study design, data collection and analysis, decision to publish, or preparation of the manuscript.

**Competing interests:** The authors have declared that no competing interests exist.

## Author summary

Lung adenocarcinoma is the predominant form of lung cancer, and for early-stage patients, surgical decisions hinge on accurately predicting tumor invasiveness. Distinguishing between non-invasive lesions suitable for limited resection and invasive tumors requiring lobectomy remains challenging with standard subjective CT interpretation. To address this, we developed a quantitative framework that analyzes the internal "texture" and structural complexity of lung nodules. A key innovation of our study is the introduction of a "3D intratumoral heterogeneity score," which uses advanced topological analysis to map the spatial connectivity and fragmentation of tumor tissue across the entire volume, rather than just a single 2D slice. We integrated these scores into a machine learning model and validated its performance on a large cohort of 1,238 patients from three different medical centers. Our results confirm that this 3D approach significantly outperforms traditional methods in identifying invasive cancer. This non-invasive, robust tool offers clinicians a powerful objective metric to guide personalized surgical planning, helping to avoid overtreatment and preserve vital lung function for patients.

## Introduction

Lung cancer remains the leading global cause of cancer-related mortality, with adenocarcinoma accounting for over 40% of all cases [1,2]. The revised WHO classification stratifies lung adenocarcinoma (LUAD) precursors and early-stage lesions into progressive biological states: atypical adenomatous hyperplasia (AAH), adenocarcinoma in situ (AIS), minimally invasive adenocarcinoma (MIA), and invasive adenocarcinoma (IAC) [3]. Critically, AIS and MIA show exceptional outcomes, with a 10-year disease-free survival rate approaching 100% after complete resection [4]. In contrast, IAC exhibits a significantly reduced 5-year survival rate of 89% and a higher risk of recurrence [5]. This prognostic dichotomy necessitates individualized surgical strategies: sublobar resection (wedge or segmentectomy) is sufficient for preinvasive lesions (AAH, AIS, MIA) [6], while IAC requires radical lobectomy accompanied by systematic lymph node dissection to mitigate metastatic potential [7–9]. Consequently, accurate preoperative assessment of pathological invasiveness directly influences therapeutic decision-making.

The invasiveness of LUAD is intrinsically linked to intratumoral heterogeneity (ITH), which manifests as spatial variations in cellular composition, metabolic activity, and microenvironmental remodeling [10]. Computed tomography (CT) serves as a critical non-invasive tool for LUAD diagnosis and staging, capturing comprehensive morphological and textural information that may reflect underlying ITH. Distinct local patterns (e.g., necrosis, angiogenesis) or global patterns (e.g., density gradients) on CT images can arise from heterogeneous tumor biology [11]. Currently, clinical practice relies on the subjective evaluation of clinicoradiologic (CR) features, such as nodule size, CT density, lobulation sign, and patient age. However, these

methodologies encounter significant limitations, including poor repeatability and a dependence on the interpreter's level of experience [12].

Presently, CT-based quantification of ITH primarily relies on two paradigms, yet both often overlook the spatial topology of heterogeneous tissues. The conventional radiomics approach utilizes single or composite features (e.g., entropy, wavelet textures) as invasiveness biomarkers [13–15]; however, it typically assumes a uniform distribution of heterogeneity, failing to quantify localized pathological variations. Conversely, habitat analysis segments tumor subregions into distinct risk profiles but frequently neglects the topological organization—specifically the connectivity and spatial arrangement—of these habitats [16,17].

To bridge these gaps, recent advancements have introduced the "ITH score," a metric originally proposed by Li et al. [18] that integrates local radiomics features with global pixel distribution patterns through unsupervised clustering. Crucially, this approach conceptualizes heterogeneity not merely as statistical variance, but as the fragmentation of topologically distinct phenotypic subregions. This methodology has demonstrated significant clinical utility in predicting pathological subtypes [19] and invasiveness in patients with LUAD [20–22]. Nevertheless, these methodologies are predominantly restricted to the largest cross-sectional CT slice operationally defined as the 2D ITH score which limits the comprehensive interpretation of subtle, volumetric heterogeneity signatures. Addressing this limitation, Zuo et al. [23] recently advanced the paradigm by introducing a "3D ITH score," calculated from the entire tumor volume to provide a more holistic characterization of heterogeneity.

Building upon these foundational studies, the present research proposes a comprehensive framework that integrates topologically distinct 2D and 3D ITH scores with standard CR features. By explicitly quantifying the volumetric connectivity and spatial complexity of tumor subregions, this study aims to capture depth-wise invasive features that planar analysis may miss. Leveraging a large-scale multicenter cohort to ensure model robustness and generalizability, the primary objective of this study is to establish these topology-aware ITH scores as pivotal diagnostic biomarkers for the preoperative prediction of LUAD invasiveness, ultimately providing evidence-based guidance for surgical decision-making.

## Materials and methods

### Patient enrollment

We consecutively enrolled patients with histopathologically confirmed LUAD who underwent surgical resection and preoperative chest CT scans between January 2018 and January 2024 at three medical centers: Xiangtan Central Hospital, the Third Xiangya Hospital of Central South University, and the Affiliated Hospital of Southwest Medical University. Inclusion criteria included: (i) histopathological confirmation of lung adenocarcinoma with available surgical specimens; (ii) a preoperative thin-section CT scan (slice thickness ≤ 1.5 mm) performed within 6 months before surgery; and (iii) lung nodules measuring 5-30 mm in maximum diameter, radiologically classified as clinical stage I (T1N0M0) according to the 9th TNM staging system [24]. Exclusion criteria included: (i) suboptimal CT image quality; (ii) presence of synchronous multiple lung adenocarcinomas or metastatic lesions; (iii) receipt of preoperative chemoradiotherapy; and (iv) unsuccessful computational processing (e.g., segmentation failures, clustering errors, or feature extraction abnormalities). A comprehensive enrollment schematic can be found in Fig 1.

This multicenter retrospective study was approved by the Institutional Review Boards of Xiangtan Central Hospital (No. 2021-07-009), The Third Xiangya Hospital of Central South University (No. K25083), and the Affiliated Hospital of Southwest Medical University (No. KY2020147), in accordance with the 2013 revision of the Declaration of Helsinki [25]. Informed consent was waived due to the retrospective nature of the study and the use of de-identified patient information.

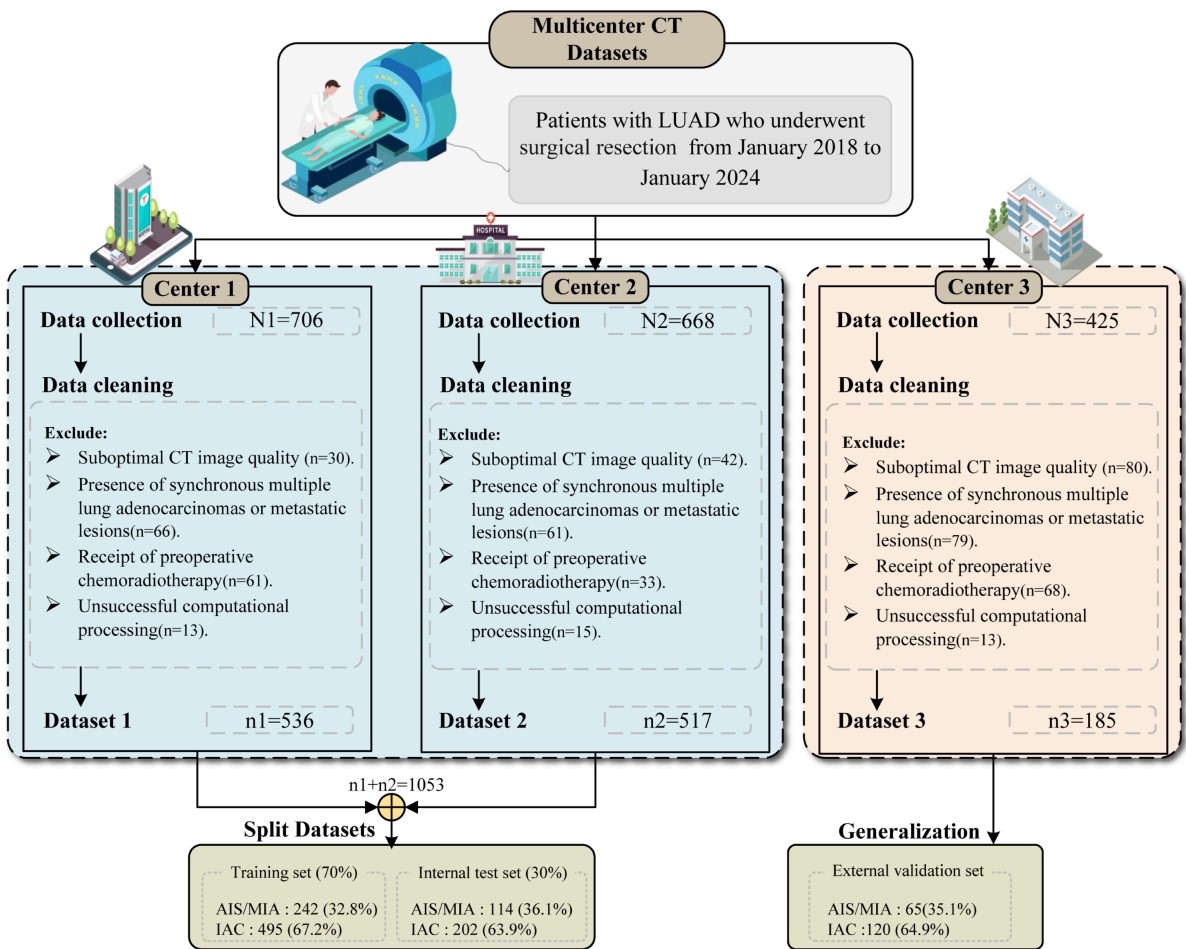

**Fig 1**. **Flowchart of patient screening and cohort stratification.** A total of 1,238 eligible patients from three medical centers were consecutively enrolled. Following rigorous exclusion criteria, patients were stratified into a training set (n=737), an internal test set (n=316), and an independent external validation set (n=185) to ensure robust model development and validation.

## Multicenter CT dataset construction and nodule segmentation

This multicenter retrospective analysis employed standardized thin-section CT protocols across all participating institutions. Scanners from multidetector CT systems were utilized, with a reconstructed slice thickness set to $\leq 1.5\,\text{mm}$. Detailed specifications of the CT acquisition protocols are provided in S1 Appendix.

All DICOM images underwent nodule segmentation using ITK-SNAP (v3.6.0) following a rigorous two-stage workflow to generate the 3D masks (denoted as $M$). The initial stage involved manual contouring of volumetric boundaries on multiplanar reconstructions, performed under fixed lung window settings (window level –600 Hounsfield units [HU], width 1500 HU) by a board-certified cardiothoracic radiologist with 5 years of subspecialty experience. This was followed by a careful review and refinement conducted by an attending thoracic radiologist with 10 years of dedicated experience to ensure the accuracy of the delineation.

## Calculation of 2D/3D ITH scores

To quantify the internal heterogeneity of LUAD nodules, we developed a dual-scale computational framework that transforms local radiomic features into topologically distinct tissue subregions. This approach extends established

methodologies [18–23] by implementing a voxel-based topology-aware quantification strategy. The workflow generates two complementary metrics: a 2D ITH score that maps planar spatial patterns on the largest cross-sectional slice, and a 3D ITH score that captures volumetric connectivity across the entire tumor mask. The complete computational procedure is formalized in Algorithm 1.

**Algorithm 1 Calculation of topologically distinct 2D and 3D ITH scores.**

**Require:** CT Image, Tumor Mask $\Omega$, Mode $M \in \{2D, 3D\}$, Initialize accumulator $\Sigma \leftarrow 0$, feature number $N \leftarrow 104$, cluster Number $K \leftarrow 6$

**Ensure:** ITH Score $\in [0, 1]$

1: **Phase 1: Initialization**
2: **if** $M = 2D$ **then**
3:    $\Omega' \leftarrow$ Largest cross-sectional axial slice of $\Omega$
4:    $w \leftarrow 2 \times 2$ (Sliding window kernel)
5:    $\mathcal{T} \leftarrow$ 8-connectivity (Pixel topology)
6: **else if** $M = 3D$ **then**
7:    $\Omega' \leftarrow \Omega$ (Full volumetric mask)
8:    $w \leftarrow 2 \times 2 \times 2$ (Sliding window kernel)
9:    $\mathcal{T} \leftarrow$ 26-connectivity (Voxel topology)
10: **end if**
11: **Phase 2: Feature Extraction & Clustering**
12: **for** each $x \in \Omega'$ **do**
13:    Extract generic feature vector $f(x)$ using kernel $w$
14: **end for**
15: Apply K-means clustering on feature matrix $F(x) = [f(x_1), \dots, f(x_N)]$ to generate cluster label $L(x)$
16: Generate Cluster Map $C$ by mapping labels $L(x) \in \{1, \dots, K\}$ to spatial coordinates
17: **Phase 3: Topology-Aware Quantification**
18: **if** $M = 2D$ **then**
19:    Calculate the total area of $\Omega'$ by $S_{total}$
20: **else if** $M = 3D$ **then**
21:    Calculate the total volume of $\Omega'$ by $V_{total}$
22: **end if**
23: **for** cluster $i = 1$ to $K$ **do**
24:    Extract binary subregion $C_i = \{x \mid L(x) = i\}$
25:    Identify connected components in $C_i$ by $\mathcal{T}$
26:    **if** $M = 2D$ **then**
27:       $n_i \leftarrow$ Count of connected components
28:       $S_{i,max} \leftarrow$ the largest connected component
29:       $\Sigma \leftarrow \Sigma + \frac{S_{i,max}}{n_i}$
30:    **else if** $M = 3D$ **then**
31:       $m_i \leftarrow$ Count of connected components
32:       $V_{i,max} \leftarrow$ the largest connected component
33:       $\Sigma \leftarrow \Sigma + \frac{V_{i,max}}{m_i}$
34:    **end if**
35: **end for**
36: **Phase 4: ITH Score Derivation**
37: **if** $M = 2D$ **then**
38:    2D ITH Score $= 1 - \frac{1}{S_{total}} \sum_{i=1}^{K} \frac{S_{i,max}}{n_i}$
39: **else if** $M = 3D$ **then**
40:    3D ITH Score $= 1 - \frac{1}{V_{total}} \sum_{i=1}^{K} \frac{V_{i,max}}{m_i}$
41: **end if**
42: **return** 2D and 3D ITH Scores

**2D ITH score derivation.** The 2D ITH score quantifies heterogeneity within the representative axial plane by analyzing the spatial connectivity patterns of tissue phenotypes. Radiomic features were extracted using a $2 \times 2$ sliding window, and pixels were partitioned into $K = 6$ topologically distinct subregions. The final score integrates the spatial dispersion of these clusters using an area-weighted formulation, as defined in Eq 1:

$$\text{2D ITH Score} = 1 - \frac{1}{S_{\text{total}}} \sum_{i=1}^{K} \frac{S_{i,\max}}{n_i}, \tag{1}$$

where $K$ represents the total number of clusters, $n_i$ denotes the number of disconnected topological components for cluster $i$, $S_{i,\max}$ is the area of the largest connected component within that cluster, and $S_{\text{total}}$ is the total area of the tumor on the largest cross-sectional slice.

**3D ITH score derivation.** The 3D ITH score extends this analysis to the volumetric domain to capture anisotropic texture variations and volumetric fragmentation along the z-axis. A $2 \times 2 \times 2$ sliding window was employed for voxel-wise feature extraction. To preserve the continuity of complex infiltrative patterns in 3D space, a 26-connectivity topology (encompassing face, edge, and corner adjacency) was utilized to identify connected voxel components. The volumetric score is calculated using Eq 2:

$$\text{3D ITH Score} = 1 - \frac{1}{V_{\text{total}}} \sum_{i=1}^{K} \frac{V_{i,\max}}{m_i}, \tag{2}$$

where $m_i$ represents the count of distinct connected volumes for cluster $i$, $V_{i,\max}$ denotes the volume of the largest connected component for that cluster, and $V_{\text{total}}$ represents the total tumor volume.

## Clinicoradiologic feature acquisition

The evaluation of CR features was conducted independently by two senior radiologists, each with over 10 years of thoracic imaging experience, who were blinded to histopathological outcomes. To resolve any inter-observer discrepancies, a consensus was reached through consultation with a third radiologist. The CR profile comprised demographic variables (age, sex) and a comprehensive set of morphological descriptors derived from thin-section CT images. Nodule size was quantified as the maximum diameter on the largest axial cross-section. Nodule attenuation (CT density) was classified into three distinct categories: pure ground-glass nodules (pGGN), part-solid nodules (PSN), and solid nodules (SN). Furthermore, specific qualitative morphological signs were systematically assessed: lobulation was defined as a scalloped or wavy margin; spiculation as linear strands extending from the nodule into the lung parenchyma; vascular convergence as the convergence of vessel structures toward the tumor; the vacuole sign as bubble-like lucencies of <5 mm within the nodule; and pleural indentation as the retraction of the visceral pleura toward the lesion. Representative examples of these features are visually detailed in Fig 2.

## Machine learning framework

**Machine learning model development.** We implemented a comprehensive supervised learning pipeline utilizing the `scikit-learn` library [26]. Six advanced machine learning algorithms were constructed: Gradient Boosting Decision Tree (GBDT), Adaptive Boosting (AdaBoost), Extreme Gradient Boosting (XGBoost), Light Gradient Boosting Machine (LightGBM), Categorical Boosting (CatBoost), and Random Forest (RF). Hyperparameter optimization was rigorously conducted within the training cohort using a grid search strategy embedded within a 5-fold cross-validation (CV). For each parameter combination, the model was trained on four folds and validated on the remaining hold-out fold to generate out-of-fold (OOF) predictions. The optimal hyperparameter configuration for each classifier was identified by maximizing the mean area under the curve (AUC) calculated across the five OOF validation sets [27].

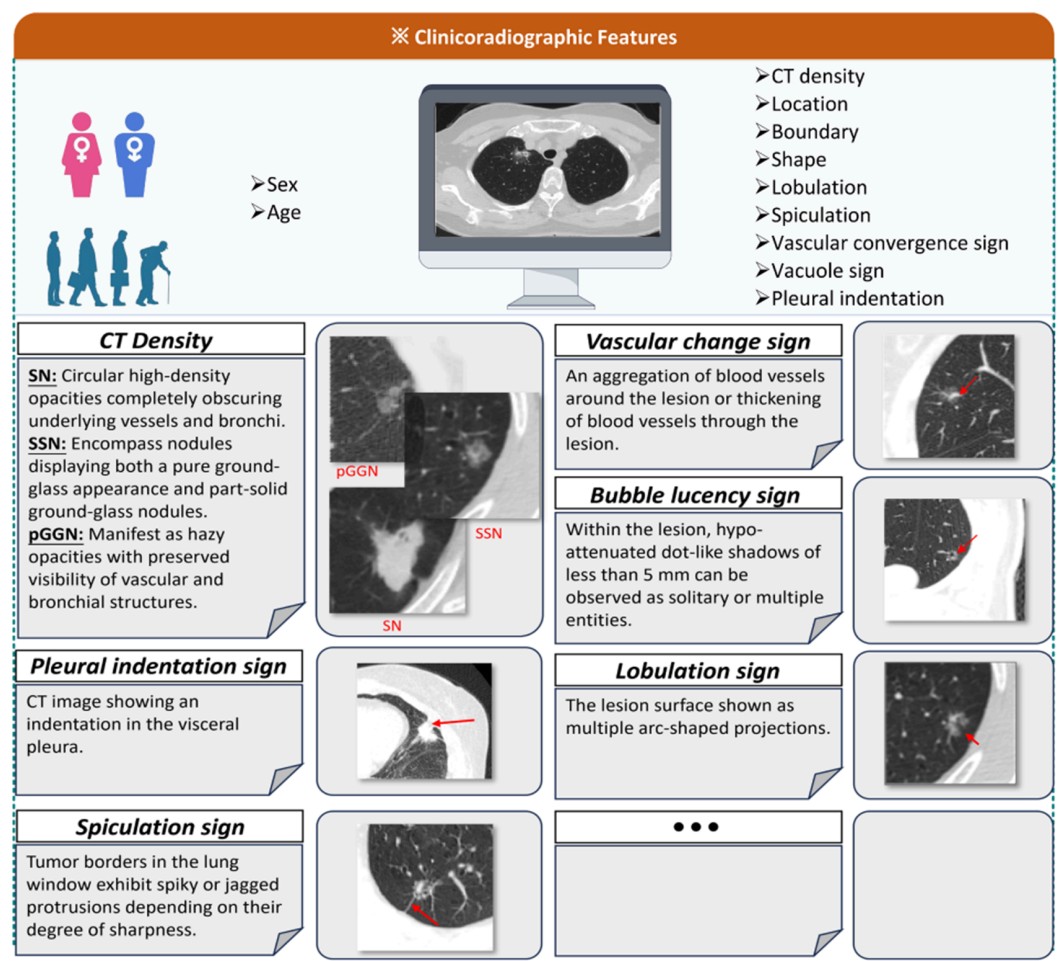

**Fig 2**. **Visual representation and definitions of clinicoradiologic (CR) features.** The figure illustrates the key morphological descriptors assessed in this study.

**Performance metrics and model selection strategy.** To ensure clinical robustness, model performance was comprehensively evaluated using the AUC as the primary discriminative metric. Complementary metrics—including accuracy, precision, recall, and F1 score were also calculated to assess classification balance. The "optimal model" was initially identified as the classifier achieving the highest AUC across the internal test set and independent external validation set, and subsequently validated through the incremental ablation study described below [22,28]. The overall design workflow of this study is illustrated in Fig 3.

**Model interpretability and incremental feature optimization.** To thoroughly evaluate model robustness and identify the most parsimonious predictive signature, we implemented a Shapley Additive Explanations (SHAP)-guided incremental feature ablation strategy across all six machine learning classifiers. First, we utilized TreeSHAP to quantify the contribution of each feature. The SHAP value $\phi_i$ represents the marginal contribution of feature $i$, calculated as:

$$\phi_i = \sum_{\mathcal{K} \subseteq F \setminus \{i\}} \frac{|\mathcal{K}|!(|F| - |\mathcal{K}| - 1)!}{|F|!} [f(\mathcal{K} \cup \{i\}) - f(\mathcal{K})] \tag{3}$$

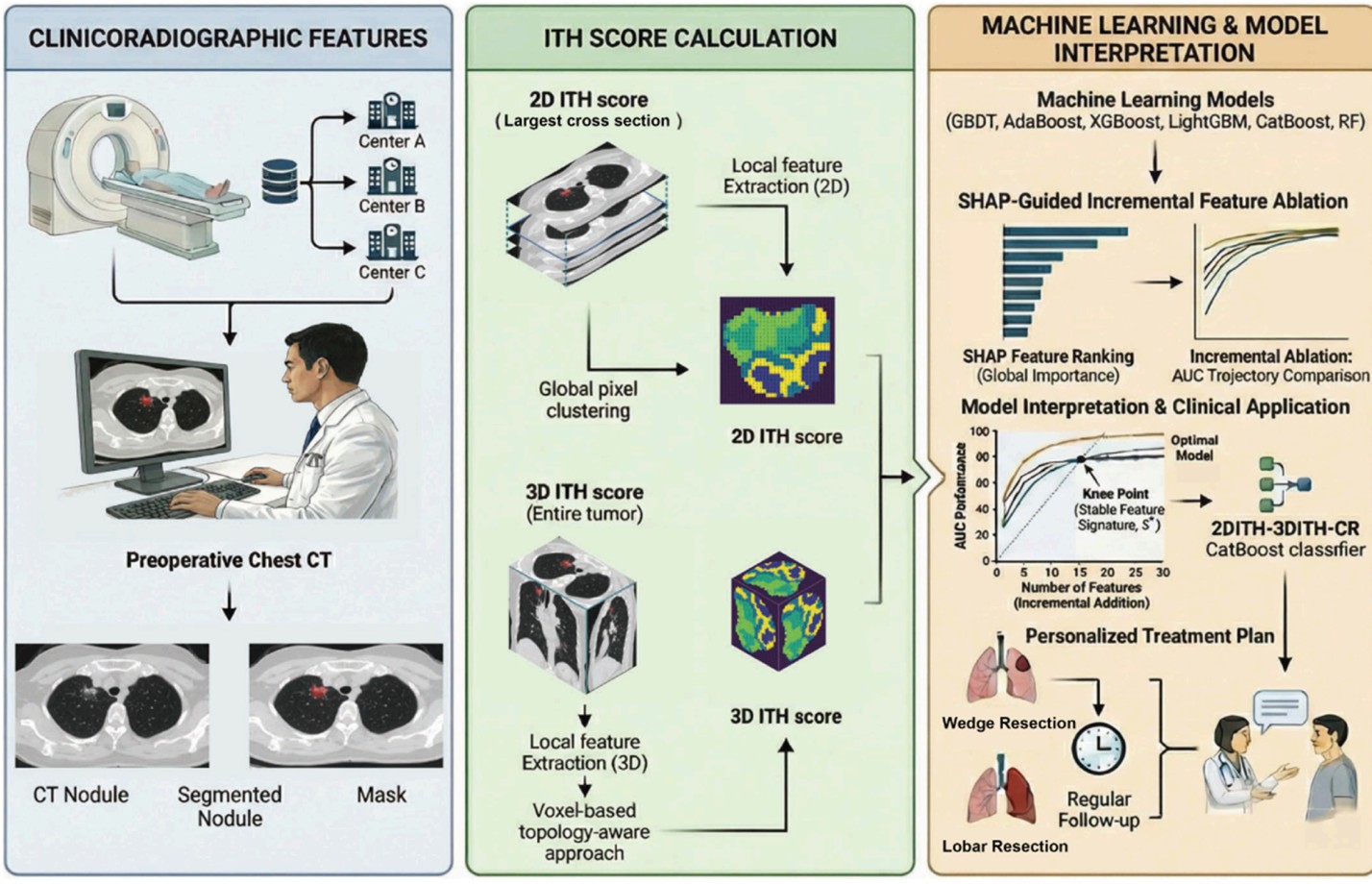

**Fig 3. Comprehensive schematic of the study design.** The workflow encompasses three primary stages: (1) Acquisition of clinicoradiologic features and calculation of dual-scale (2D and 3D) intratumoral heterogeneity (ITH) scores; (2) Development and optimization of six machine learning classifiers (GBDT, AdaBoost, XGBoost, LightGBM, CatBoost, and RF); and (3) Model interpretation using SHAP values alongside rigorous performance evaluation across multicenter datasets.

where $F$ is the set of all input features, $\mathcal{K}$ represents a subset of features excluding feature $i$, and $f$ denotes the prediction function of the machine learning model.

For each classifier, the constituent features were ranked in descending order based on their global importance score ($I_i$), defined as the mean absolute SHAP value across the cohort ($I_i = \frac{1}{N}\sum_{j=1}^{N}|\phi_i^{(j)}|$), where $N$ is the total number of samples and $\phi_i^{(j)}$ is the SHAP value of feature $i$ for sample $j$.

Based on these rankings, we executed a sequential forward selection process for every classifier (Algorithm 2). This comprehensive analysis served two critical purposes. First, regarding model selection, we compared the AUC trajectories of all classifiers during the iterative addition of features; the classifier exhibiting consistently superior performance curves was identified as the best model ($\mathcal{C}_{opt}$). Second, for feature optimization, we pinpointed the "knee point" within this optimal model—specifically, the minimal feature subset size $I^*$ where the marginal performance gain ($\Delta$) dropped below a negligible threshold ($\varepsilon$)—thereby establishing the final stable feature signature, formally denoted as the optimal subset $S^*$.

**Algorithm 2 Multi-model SHAP-guided iterative feature elimination.**

**Require:** Set of Classifiers $\mathbb{C}$ (GBDT, CatBoost, etc.), Dataset $\mathcal{D}$, Full Features $F$, feature number $N$
**Ensure:** best classifier $\mathcal{C}_{opt}$, optimal Subset $S^*$

1: **Phase 1: Performance Trajectory Generation**
2: **for** each classifier $c \in \mathbb{C}$ **do**
3: Train $c$ using all features $F$
4: Rank features $F_R = \{f^{(1)}, ..., f^{(N)}\}$ by Global SHAP Importance
5: **for** $i = 1$ to $N$ **do**
6: Form subset $F_i = \{f^{(1)}, ..., f^{(i)}\}$ from $F_R$
7: Train classifier $c$ with $F_i$ using 5-fold CV
8: Record performance metric $AUC_{c,i}$
9: **end for**
10: **end for**
11: **Phase 2: Comparative Selection & Optimization**
12: $\mathcal{C}_{opt} \leftarrow$ Select classifier with consistently highest $AUC$ trajectory
13: **Phase 3: Knee Point Identification (for $\mathcal{C}_{opt}$)**
14: Initialize $l \leftarrow 1$, Threshold $\varepsilon \approx 0$
15: **while** $l < N$ **do**
16: $\Delta \leftarrow AUC_{\mathcal{C}_{opt}, l+1} - AUC_{\mathcal{C}_{opt}, l}$
17: **if** $\Delta \leq \varepsilon$ **then** ▷ Marginal gain negligible
18: $l^* \leftarrow l$
19: **break**
20: **else**
21: $l \leftarrow l + 1$
22: **end if**
23: **end while**
24: **return** $\mathcal{C}_{opt}$, $S^* = \{f^{(1)}, ..., f^{(l^*)}\}$

## Results

### Patient characteristics

A total of 1,238 eligible patients were recruited from three medical centers. A cohort of 1,053 consecutively recruited patients from Centers 1 and 2 was randomly allocated via stratified sampling into training (n=737) and internal test (n=316) sets, following a 7:3 ratio to ensure consistent class distributions. Additionally, 185 prospectively enrolled patients from Center 3 comprised an independent external validation set. Within the training set, 737 patients were included, among whom 495 (67.2%) were diagnosed with IAC. The internal test cohort consisted of 316 patients, with 202 (63.9%) diagnosed with IAC. The external validation set included 185 patients, of whom 120 (64.9%) were diagnosed with IAC.

The detailed characteristics of the enrolled patients are presented in Table 1. No statistically significant differences were found among the three groups, as all $p$-values were greater than 0.05, indicating that the groups were comparable.

### 2D/3D ITH scores and pathological invasiveness

The distributions of 2D and 3D ITH scores across the training, internal test, and external validation sets are presented in Fig 4. In the context of clinical stage I LUAD, lesions classified as IAC demonstrated significantly higher 2D and 3D ITH scores compared with the AIS/MIA group across all cohorts ($p < 0.001$). These findings indicate that 2D/3D ITH score derived from preoperative CT images enables significant stratification between invasive and pre-/minimally invasive histological subtypes.

**Table 1**. Comparative analysis of clinicoradiologic features across training, internal test, and external validation sets.

| Variables | Total (n = 1238) | Training set (n = 737) | Internal Test (n = 316) | External Validation (n = 185) | P value |
|---|---|---|---|---|---|
| **Pathological diagnosis, n (%)** | | | | | 0.56 |
| AIS/MIA | 421 (34) | 242 (32.8) | 114 (36.1) | 65 (35.1) | |
| IAC | 817 (66) | 495 (67.2) | 202 (63.9) | 120 (64.9) | |
| **Sex, n (%)** | | | | | 0.131 |
| Female | 821 (66.3) | 479 (65) | 224 (70.9) | 118 (63.8) | |
| Male | 417 (33.7) | 258 (35) | 92 (29.1) | 67 (36.2) | |
| **Age (y), Median (IQR)** | 57 (51, 66) | 58 (50, 66) | 57 (52, 65) | 58 (49, 66) | 0.896 |
| **Nodule size (mm), Median (IQR)** | 15.8 (11.7, 20.6) | 16.2 (11.7, 20.6) | 15.6 (11.7, 20.6) | 15.3 (11.2, 20.6) | 0.734 |
| **CT Density, n (%)** | | | | | 0.578 |
| pGGN | 669 (54) | 402 (54.5) | 166 (52.5) | 101 (54.6) | |
| PSN | 321 (25.9) | 198 (26.9) | 80 (25.3) | 43 (23.2) | |
| SN | 248 (20) | 137 (18.6) | 70 (22.2) | 41 (22.2) | |
| **Location, n (%)** | | | | | 0.936 |
| RUL | 438 (35.4) | 264 (35.8) | 112 (35.4) | 62 (33.5) | |
| RML | 84 (6.8) | 49 (6.6) | 19 (6) | 16 (8.6) | |
| RLL | 209 (16.9) | 118 (16) | 58 (18.4) | 33 (17.8) | |
| LUL | 340 (27.5) | 208 (28.2) | 85 (26.9) | 47 (25.4) | |
| LLL | 167 (13.5) | 98 (13.3) | 42 (13.3) | 27 (14.6) | |
| **Margin, n (%)** | | | | | 0.729 |
| Well-defined | 930 (75.1) | 549 (74.5) | 238 (75.3) | 143 (77.3) | |
| Ill-defined | 308 (24.9) | 188 (25.5) | 78 (24.7) | 42 (22.7) | |
| **Lobulation sign, n (%)** | | | | | 0.536 |
| Absent | 573 (46.3) | 346 (46.9) | 138 (43.7) | 89 (48.1) | |
| Present | 665 (53.7) | 391 (53.1) | 178 (56.3) | 96 (51.9) | |
| **Spiculation sign, n (%)** | | | | | 0.86 |
| Absent | 673 (54.4) | 398 (54) | 171 (54.1) | 104 (56.2) | |
| Present | 565 (45.6) | 339 (46) | 145 (45.9) | 81 (43.8) | |
| **Vascular convergence sign, n (%)** | | | | | 0.082 |
| Absent | 281 (22.7) | 153 (20.8) | 76 (24.1) | 52 (28.1) | |
| Present | 957 (77.3) | 584 (79.2) | 240 (75.9) | 133 (71.9) | |
| **Vacuole sign, n (%)** | | | | | 0.434 |
| Absent | 1004 (81.1) | 591 (80.2) | 264 (83.5) | 149 (80.5) | |
| Present | 234 (18.9) | 146 (19.8) | 52 (16.5) | 36 (19.5) | |
| **Pleural indentation sign, n (%)** | | | | | 0.863 |
| Absent | 524 (42.3) | 315 (42.7) | 134 (42.4) | 75 (40.5) | |
| Present | 714 (57.7) | 422 (57.3) | 182 (57.6) | 110 (59.5) | |
| **Shape, n (%)** | | | | | 0.647 |
| Absent | 670 (54.1) | 391 (53.1) | 177 (56) | 102 (55.1) | |
| Present | 568 (45.9) | 346 (46.9) | 139 (44) | 83 (44.9) | |

**Abbreviations**: pGGN, pure ground-glass nodule; PSN, part-solid nodule; SN, solid nodule; AIS, adenocarcinoma in situ; MIA, minimally invasive adenocarcinoma; IAC, invasive adenocarcinoma; RUL, right upper lobe; RLL, right lower lobe; RML, right middle lobe; LUL, left upper lobe; LLL, left lower lobe; IQR, interquartile range.

## Performance evaluation and optimal model selection

The ROC curves for distinguishing IAC from AIS/MIA across the six machine learning classifiers (GBDT, AdaBoost, XGBoost, LightGBM, CatBoost, and RF) are illustrated in Fig 5. All models incorporated CR features combined with 2D and 3D ITH scores as input predictors.

Strictly adhering to the selection criteria defined in the Methods, we identified the "optimal model" based on the highest AUC performance across both the internal test and independent external validation sets. Among the evaluated classifiers, CatBoost emerged as the top performer, achieving an AUC of 0.867 in the internal test set and 0.881 in the external validation set. As summarized in Table 2, CatBoost also demonstrated superior balance across complementary metrics,

PLOS Digital Health

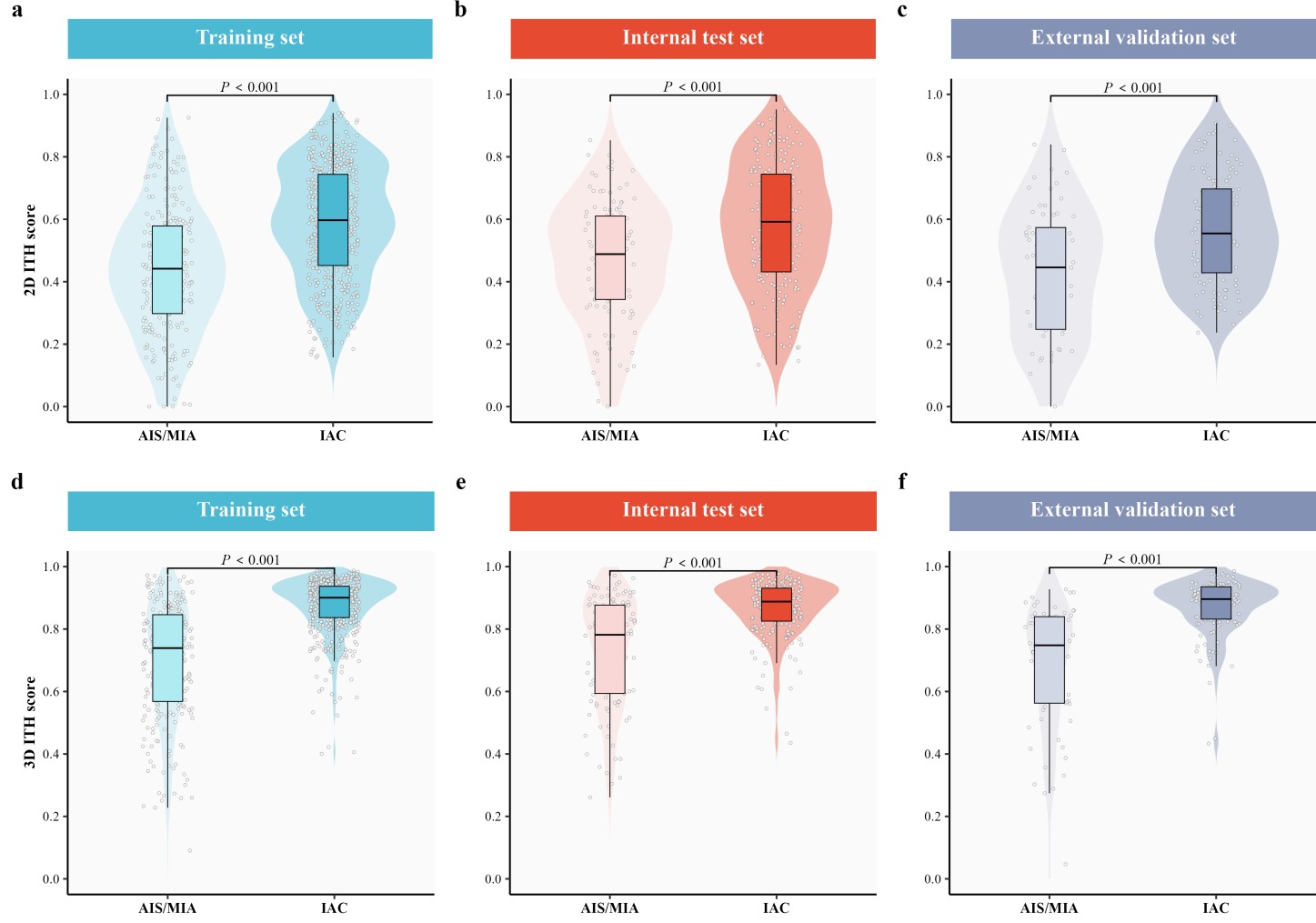

**Fig 4**. **Comparative analysis of 2D and 3D ITH scores across cohorts.** The violin plots display the distribution of heterogeneity scores across the training (a, d), internal test (b, e), and external validation sets (c, f). Statistical analysis reveals that invasive adenocarcinoma (IAC) lesions exhibit significantly higher 2D and 3D ITH scores compared to Adenocarcinoma in situ/minimally invasive adenocarcinoma (AIS/MIA) across all datasets ($P < 0.001$), validating the discriminatory power of these quantitative metrics.

including accuracy, precision, recall, and F1 score. Consequently, we established this optimal configuration—integrating 2D ITH scores, 3D ITH scores, and CR features—as the 2DITH-3DITH-CR CatBoost classifier for the identification of IAC.

**Model interpretation and optimal signature identification.** The execution of the multi-model SHAP-Guided Iterative Feature Elimination framework yielded distinctive performance trajectories for the six machine learning classifiers, as visualized in Fig 6. During the trajectory generation stage, all classifiers exhibited rapid performance gains with the initial accumulation of high-importance features. In alignment with our comparative selection criteria, the CatBoost classifier (red line) demonstrated a consistently superior AUC trajectory across the majority of iterations, thereby validating it as the best classifier ($\mathcal{C}_{opt}$) for clinical implementation. Furthermore, consistent with the knee point identification logic, a decisive inflection point was observed at a feature subset size of $l^* = 5$ (marked by the vertical dashed line). At this threshold, the marginal gain in AUC ($\Delta$) dropped below a negligible threshold ($\varepsilon$), indicating that the top five features–anchored by

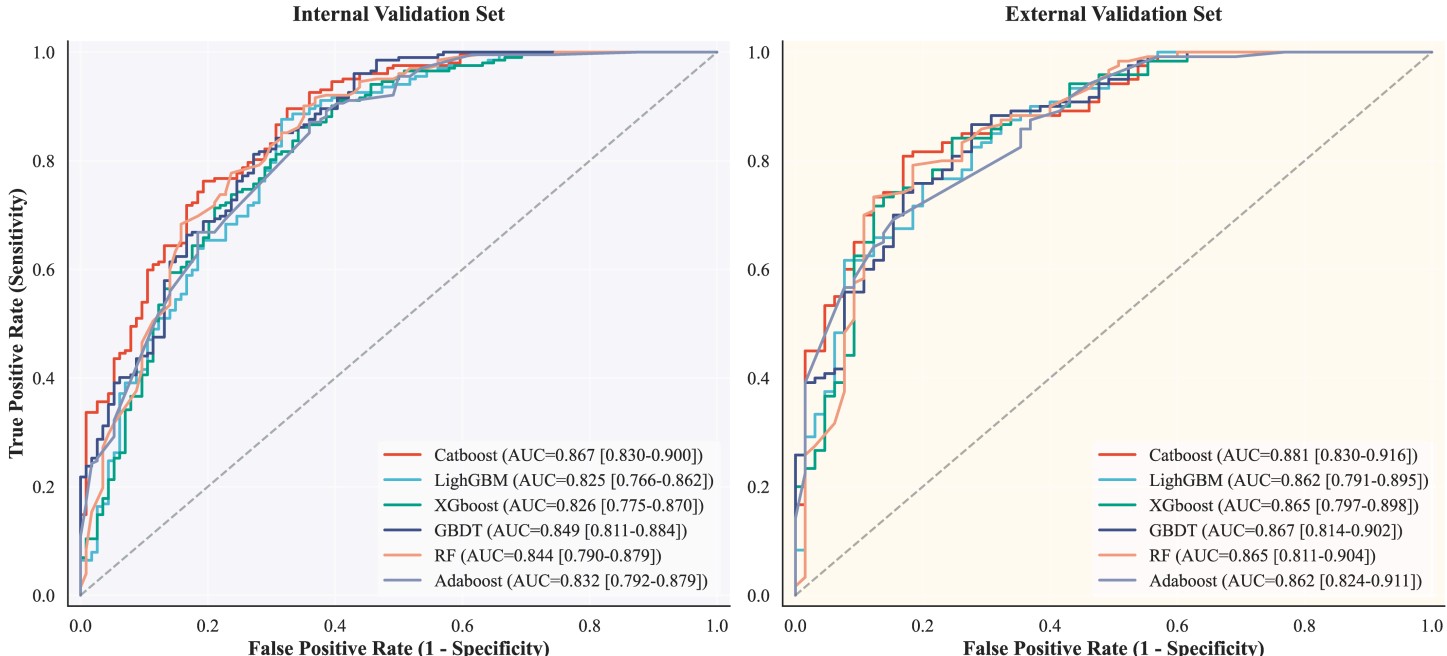

**Fig 5**. **Receiver Operating Characteristic (ROC) curves for invasiveness prediction.** The curves illustrate the diagnostic performance of six machine learning classifiers in distinguishing IAC from AIS/MIA within the internal test set (Left) and the independent external validation set (Right). The 2DITH-3DITH-CR CatBoost model demonstrates superior efficacy, achieving the highest Area Under the Curve (AUC) in both internal (0.867) and external (0.881) validations.

**Table 2**. **Assessment of diagnostic efficacy across diverse machine learning approaches.**

| Model | AUC | Accuracy | Precision | Recall | F1 score |
|---|---|---|---|---|---|
| Internal test set | | | | | |
| GBDT | 0.849 | 0.801 | 0.804 | 0.801 | 0.791 |
| XGBoost | 0.826 | 0.804 | 0.806 | 0.804 | 0.795 |
| LightGBM | 0.825 | 0.804 | 0.805 | 0.804 | 0.796 |
| CatBoost | **0.867** | 0.816 | 0.825 | 0.816 | 0.806 |
| AdaBoost | 0.832 | 0.791 | 0.790 | 0.791 | 0.783 |
| RF | 0.844 | 0.795 | 0.791 | 0.795 | 0.790 |
| External validation set | | | | | |
| GBDT | 0.867 | 0.816 | 0.815 | 0.816 | 0.816 |
| XGBoost | 0.865 | 0.805 | 0.803 | 0.805 | 0.803 |
| LightGBM | 0.862 | 0.800 | 0.797 | 0.800 | 0.797 |
| CatBoost | **0.881** | 0.795 | 0.792 | 0.795 | 0.793 |
| AdaBoost | 0.862 | 0.784 | 0.780 | 0.784 | 0.781 |

**Abbreviation**: AUC, area under curve; GBDT, gradient boosting decision tree; AdaBoost, adaptive boosting; XGBoost, extreme gradient boosting; LightGBM, light gradient boosting machine; CatBoost, categorical boosting; RF, random forest.

the topology-aware heterogeneity metrics—constitute the optimal subset ($S^*$) that maximizes diagnostic accuracy while minimizing model complexity.

Fig 7 provides a comprehensive interpretation of the 2DITH-3DITH-CR CatBoost classifier using SHAP analysis. Specifically, the topologically distinct 3D ITH score was identified as the most significant contributor, exhibiting the highest mean absolute SHAP value (mean $|\phi|$) and accounting for approximately 24.0% of the total predictive power. This finding

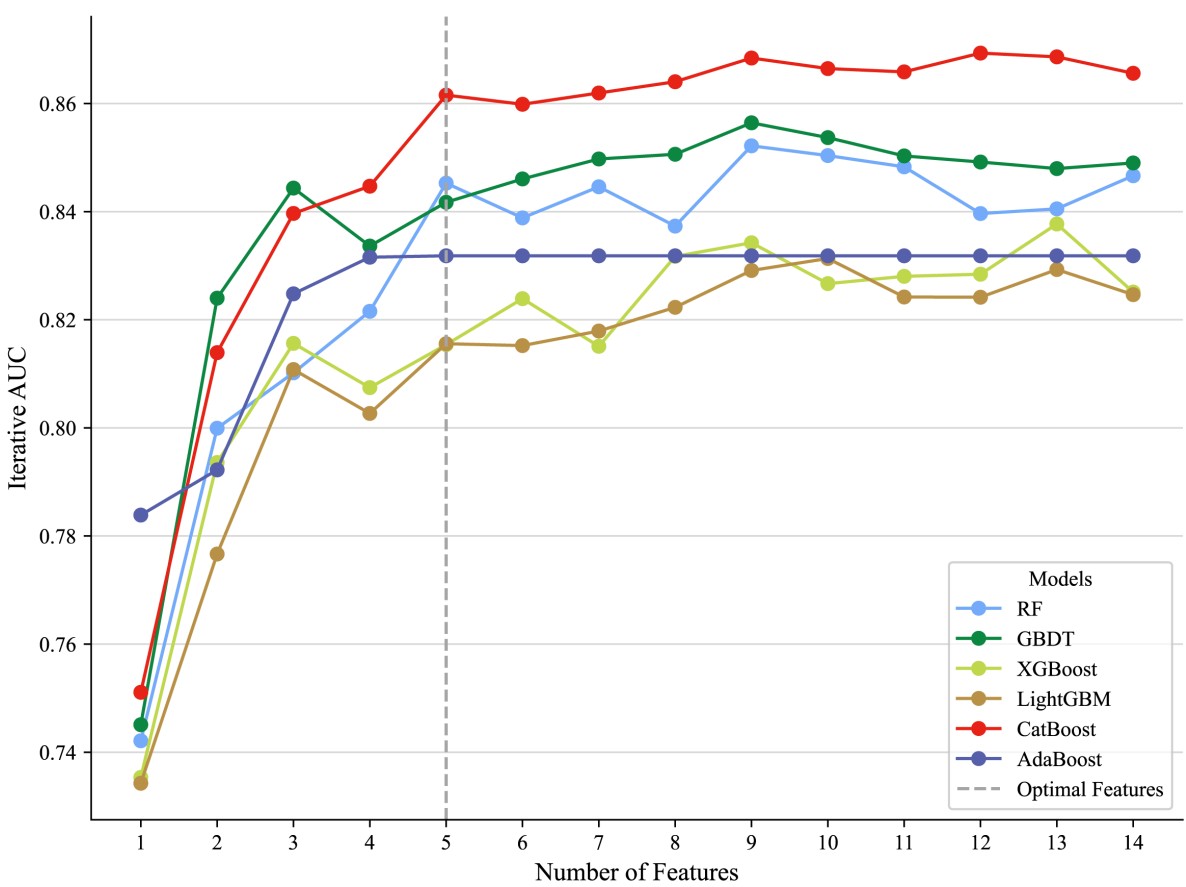

**Fig 6**. **Performance trajectories of six machine learning classifiers during SHAP-guided iterative feature elimination.** The line graph tracks the AUC evolution for Random Forest (RF), Gradient Boosting Decision Tree (GBDT), XGBoost, LightGBM, CatBoost, and AdaBoost as features are sequentially added based on their SHAP importance rankings. The vertical dashed line (Optimal Features) marks the decisive "knee point" at $l* = 5$ for the top-performing CatBoost model, identifying the minimal feature subset required to achieve maximal diagnostic accuracy while ensuring model parsimony.

underscores the superiority of volumetric topology-aware quantification over planar analysis. Corroborated by the iterative evaluation in Fig 6, the 3D ITH score proved to be the most robust feature for assessing LUAD invasiveness, followed by nodule size, CT density, 2D ITH score, and patient age.

## Feature ablation analysis

To evaluate the incremental contribution of each feature domain to the predictive capability, we conducted a systematic feature ablation study on the optimal 2DITH-3DITH-CR CatBoost classifier. Fig 8 illustrates the ROC curves representing the degradation in diagnostic performance as features were selectively removed, with detailed metrics summarized in Table 3. The full integration model (2DITH-3DITH-CR CatBoost) consistently achieved the highest AUCs across both the internal test set (AUC = 0.867) and the external validation set (AUC = 0.881), confirming that the synergistic combination of all three feature domains yields the most robust predictive signature.

Dissecting the individual contributions revealed distinct roles for each modality. The 3D ITH score exhibited the strongest generalization capability, as models relying solely on 3D ITH features significantly outperformed those relying solely on 2D ITH features (External AUC: 0.844 vs. 0.647). Concurrently, the CR features served as a critical performance

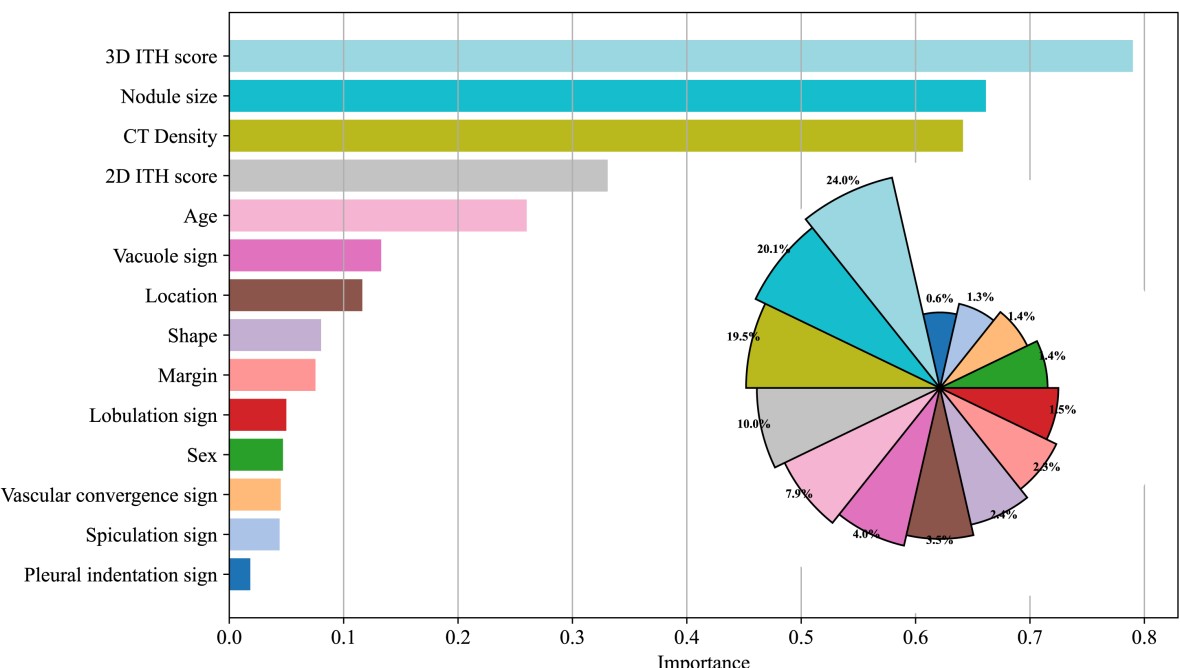

**Fig 7**. **Model interpretability analysis using SHAP values.** The bar chart ranks predictors by their mean absolute SHAP values, identifying the 3D ITH score as the most influential feature (contributing approximately 24.0% to the model's output), followed by nodule size and CT density. This hierarchy highlights the dominance of quantitative volumetric heterogeneity over qualitative morphological signs in predicting invasiveness.

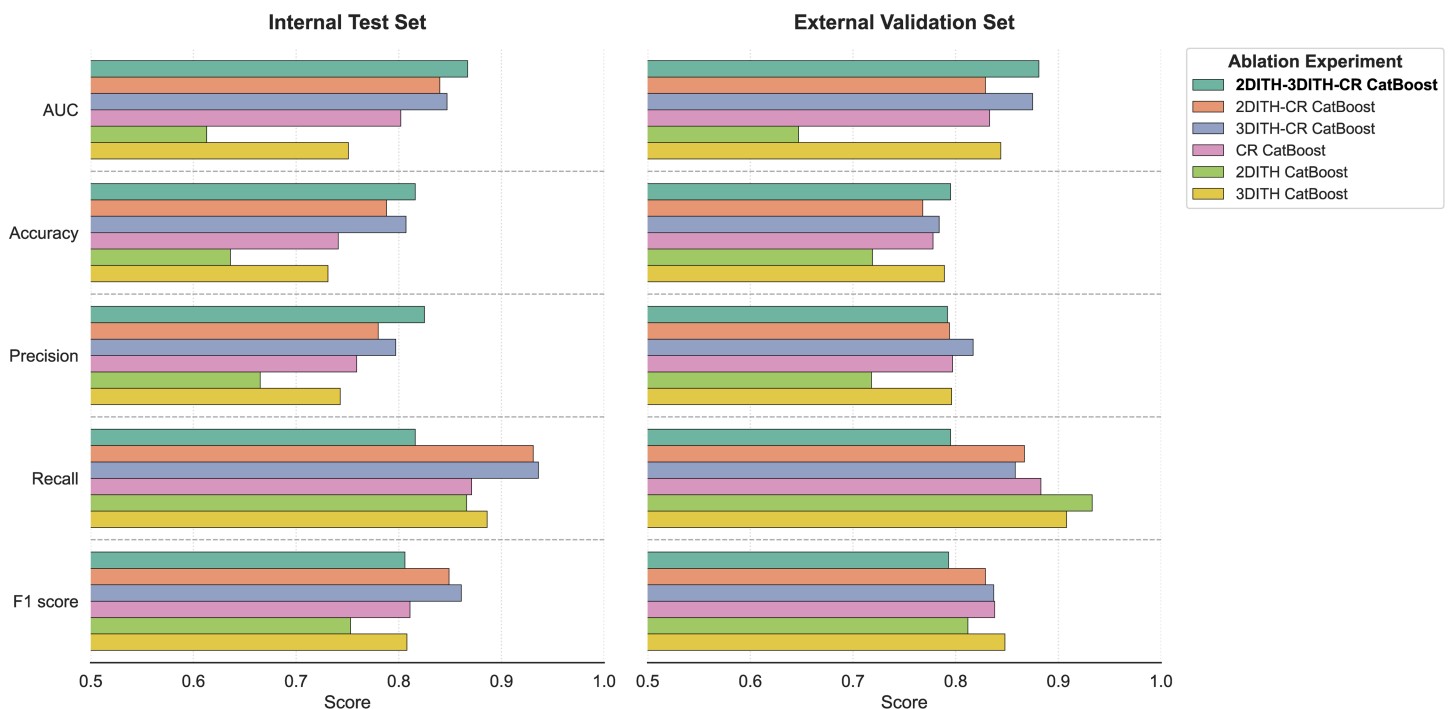

**Fig 8**. **Diagnostic performance comparison in feature ablation study.** The bar charts compare the AUC, Accuracy, Precision, Recall, and F1 scores across different model configurations in both internal and external cohorts.

**Table 3**. Diagnostic performance of feature-ablated models in internal test set and external validation set.

| Model | AUC | Accuracy | Precision | Recall | F1 score |
|---|---|---|---|---|---|
| Internal test set | | | | | |
| 2DITH-3DITH-CR CatBoost | **0.867** | 0.816 | 0.825 | 0.816 | 0.806 |
| 2DITH-CR CatBoost | 0.840 | 0.788 | 0.780 | 0.931 | 0.849 |
| 3DITH-CR CatBoost | 0.847 | 0.807 | 0.797 | 0.936 | 0.861 |
| CR CatBoost | 0.802 | 0.741 | 0.759 | 0.871 | 0.811 |
| 2DITH CatBoost | 0.613 | 0.636 | 0.665 | 0.866 | 0.753 |
| 3DITH CatBoost | 0.751 | 0.731 | 0.743 | 0.886 | 0.808 |
| External validation set | | | | | |
| 2DITH-3DITH-CR CatBoost | **0.881** | 0.795 | 0.792 | 0.795 | 0.793 |
| 2DITH-CR CatBoost | 0.829 | 0.768 | 0.794 | 0.867 | 0.829 |
| 3DITH-CR CatBoost | 0.875 | 0.784 | 0.817 | 0.858 | 0.837 |
| CR CatBoost | 0.833 | 0.778 | 0.797 | 0.883 | 0.838 |
| 2DITH CatBoost | 0.647 | 0.719 | 0.718 | 0.933 | 0.812 |
| 3DITH CatBoost | 0.844 | 0.789 | 0.796 | 0.908 | 0.848 |

**Abbreviation**: 2DITH-3DITH-CR CatBoost, CatBoost classifier with 2D and 3D intratumoral heterogeneity and clinicoradiologic features; 2DITH-CR CatBoost, CatBoost classifier with 2D intratumoral heterogeneity and clinicoradiologic features; 3DITH-CR CatBoost, CatBoost classifier with 3D intratumoral heterogeneity and clinicoradiologic features; CR CatBoost, CatBoost classifier with clinicoradiologic features; 2DITH CatBoost, CatBoost classifier with 2D intratumoral heterogeneity; 3DITH CatBoost, CatBoost classifier with 3D intratumoral heterogeneity.

baseline; even without heterogeneity metrics, the CR CatBoost model maintained a respectable performance (External AUC = 0.833). Furthermore, the inclusion of 2D ITH scores provided a fine-tuning effect to the model. While the exclusion of 2D ITH (resulting in the 3DITH-CR model) led to only a marginal decline in AUC (0.881 to 0.875 in external validation), its integration in the full model contributed to maximizing the overall diagnostic stability.

## Discussion

To our knowledge, this multicenter study is the first to establish volumetric ITH quantification—termed the 3D ITH score—as a pivotal biomarker for the preoperative prediction of pathological invasiveness in clinical stage I LUAD. By integrating this novel 3D metric with the established 2D ITH score and standard CR features, we developed a multimodal 2DITH-3DITH-CR CatBoost classifier. This model demonstrated superior discriminative performance and robustness, achieving an AUC of 0.867 in the internal test set and 0.881 in the independent external validation set. Our SHAP-guided interpretability analysis revealed that the 3D ITH score was the paramount predictive feature, followed by nodule size, CT density, 2D ITH score, and patient age. Furthermore, feature ablation experiments provided definitive evidence that the 3D ITH score serves as the primary driver of the model's generalization capability, distinguishing it as a non-redundant cornerstone for invasiveness assessment.

Intratumoral heterogeneity fundamentally originates from the spatial disorganization of diverse cellular populations and the adaptive remodeling of the tumor microenvironment. Capturing this complex biological phenomenon radiologically necessitates the concurrent quantification of both local radiomics features, which reflect variations in cellular phenotypes, and global spatial distributions, which encode architectural disruptions. Since its initial proposal by Li et al. [18], the concept of the ITH score has been progressively applied to evaluate biological characteristics across various malignancies, including breast [29,30], gastric [31], and colorectal cancers [32]. In the context of LUAD, the 2D ITH score—derived solely from the largest cross-sectional slice—has shown efficacy in predicting prognosis [18], pathological subtype [19], and invasiveness [20–22]. Consistent with these precedents, our study observed that IAC lesions exhibited significantly higher 2D ITH scores compared to AIS/MIA ($p < 0.001$), confirming that planar heterogeneity metrics effectively reflect the invasive potential of LUAD.

However, planar analysis inherently neglects the anisotropic growth patterns of tumors. Building upon the multiperspective framework established by Zuo et al. [23], our volumetric 3D ITH score addresses this limitation through the voxel-level integration of texture attributes with global distribution patterns. By utilizing a 26-connectivity topology, the 3D ITH score preserves the spatial relationships between tumor subregions along all three anatomical axes. This allows for the detection of depth-wise invasive features, such as discontinuous micrometastatic foci and gradients of necrosis-viability transitions, which may be obscured in 2D views. This theoretical advantage was empirically validated by our feature ablation study, where the 3D ITH score not only exhibited the strongest individual generalization capability (AUC = 0.844) but also significantly outperformed the planar-only approach (AUC = 0.647). Crucially, while the exclusion of 2D features yielded minimal impact, the removal of volumetric features precipitated a sharp decline in predictive accuracy, underscoring the indispensability of 3D heterogeneity in capturing the full spectrum of tumor invasiveness. Notably, the observation that our model achieved slightly higher performance in the external validation set (AUC=0.881) compared to the internal test set (AUC=0.867) addresses potential concerns regarding overfitting. Rather than indicating model bias, this superior generalization likely reflects the intrinsic stability of the 3D ITH metric. Unlike conventional radiomics that can be sensitive to scanner-specific noise, the topology-aware ITH score quantifies relative spatial connectivity ratios, making it inherently robust against the variability of CT acquisition protocols across different medical centers.

Additionally, our SHAP analysis offers critical insights into the clinical evaluation of LUAD. Among CR features, quantitative metrics—specifically nodule size and CT density—dominated the prediction of invasiveness, whereas qualitative morphological descriptors (e.g., spiculation and lobulation signs) exhibited negligible contributions. This finding challenges traditional diagnostic paradigms but aligns with emerging evidence. Meta-analyses by Yang et al. [33] and He et al. [34] have demonstrated that objective CT density metrics outperform subjective morphological signs in differentiating invasive lesions. Similarly, studies by Fu et al. [35] and Zuo et al. [36] identified nodule size as the primary independent risk factor for invasiveness. Collectively, these results underscore the limitations of subjective morphological interpretation in the era of precision oncology and support a shift towards quantitative, reproducible imaging biomarkers.

Methodologically, the superior performance of the CatBoost classifier over other machine learning algorithms (e.g., RF, SVM, LightGBM) can be attributed to its distinct algorithmic advantages tailored for heterogeneous biomedical data. Unlike traditional gradient boosting methods, CatBoost employs an "ordered boosting" scheme that effectively mitigates prediction shift and target leakage, thereby reducing the risk of overfitting on small-to-medium-sized medical datasets [37,38]. This architectural robustness likely contributed to the high stability observed in our multicenter validation. Furthermore, its native capability to process categorical features (e.g., tumor location, sex) without obligate one-hot encoding preserves the original data structure, allowing for more efficient modeling of non-linear interactions between complex radiomic signatures and clinical variables.

Despite these advancements, this study has limitations. First, the retrospective inclusion of patients with confirmed pathology introduces potential selection bias towards surgically managed cases. Future prospective studies enrolling all indeterminate nodules, including those under active surveillance, are necessary to confirm real-world applicability. Second, we did not perform a direct spatial registration between the ITH subregions and histopathological specimens. Consequently, while the 3D ITH score correlates strongly with invasiveness, the specific biological identity of these topological clusters (e.g., distinguishing necrosis from high cellularity) remains to be validated. Future research integrating spatial transcriptomics and multiplex immunohistochemistry is required to map these heterogeneity patterns against specific tissue microstructures, establishing direct radiopathologic correlates. Finally, while validated in LUAD, the generalizability of this volumetric scoring methodology to other solid tumors warrants further investigation. Clinically, this approach is highly cost-effective as it utilizes standard preoperative CT images without additional expense. However, routine implementation faces barriers, specifically the need for software integration into radiological workflows and prospective standardization of diagnostic cut-offs.

## Conclusion

This study establishes volumetric ITH quantification as a robust biomarker for preoperatively predicting pathological invasiveness in clinical stage I LUAD. The proposed 2DITH-3DITH-CR CatBoost classifier demonstrated superior diagnostic performance and generalizability across multicenter cohorts compared to traditional methods. By capturing comprehensive volumetric spatial patterns, this approach facilitates precise risk stratification, marking a significant advancement from descriptive radiology to quantitative precision oncology for optimizing early-stage LUAD management.

## Supporting information

**S1 Appendix. CT acquisition protocols.** Detailed specifications of the CT acquisition protocols used at the three participating medical centers.
(PDF)

## Acknowledgments

The authors thank the staff at the participating medical centers for their assistance with data collection. Funding, data availability, and author contributions are provided in the relevant sections of the submission system.

## Author contributions

**Conceptualization:** Zhichao Zuo, Xiaohong Fan, Wei Li, Qi Liang.

**Data curation:** Zhichao Zuo, Ying Zeng, Wanyin Qi, Wen Liu, Wei Li.

**Supervision:** Xiaohong Fan.

**Writing – original draft:** Zhichao Zuo.

**Writing – review & editing:** Xiaohong Fan, Qi Liang.

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
