## [Decision Letter · Decision Letter 0]

8 Jan 2026

PDIG-D-25-00799Enhancing Invasiveness Prediction in Clinical Stage I Lung Adenocarcinoma: A Machine Learning Approach Integrating 2D/3D Intratumoral Heterogeneity Scores with Clinicoradiologic Features from a Multicenter CT DatasetPLOS Digital Health
Dear Dr. Qi Liang,
Thank you for submitting your manuscript to PLOS Digital Health. After careful consideration, we feel that it has merit but does not fully meet PLOS Digital Health's publication criteria as it currently stands. Therefore, we invite you to submit a revised version of the manuscript that addresses the points raised during the review process.

1. Please provide an Author Summary. This should appear in your manuscript between the Abstract (if applicable) and the Introduction, and should be 150–200 words long. The aim should be to make your findings accessible to a wide audience that includes both scientists and non-scientists. Sample summaries can be found on our website under Submission Guidelines: [LINK]

https://journals.plos.org/digitalhealth/s/submission-guidelines#loc-parts-of-a-submission

If the reviewer comments include a recommendation to cite specific previously published works, please review and evaluate these publications to determine whether they are relevant and should be cited. There is no requirement to cite these works unless the editor has indicated otherwise. 
**Additional Editor Comments:**
This manuscript presents a well-designed multicenter study that integrates novel 2D and 3D intratumoral heterogeneity (ITH) scores with clinicoradiologic features to preoperatively predict invasiveness in clinical stage I lung adenocarcinoma. The study is methodologically sound, leverages a substantial dataset (n=1,238) with external validation, and achieves strong predictive performance (AUC up to 0.881). The topic is clinically relevant and aligns with the scope of PLOS Digital Health. However, several issues—primarily related to clarity, methodological detail, and interpretation—must be addressed before the manuscript can be considered for publication.

1. Methodological Clarifications:

Provide more detail on the radiomics feature extraction process;

Clarify how the optimal number of clusters (6 for 2D, 5 for 3D) was determined. Justify these choices with references or validation;

Explain the handling of missing data and class imbalance in model training.

2. Results Presentation:

In Table 1, correct the typo in “AIC/MIA” to “AIS/MIA.”?

Ensure all figures (especially Figs. 4–8) are clearly labeled and legible in the final version.

3. Discussion and Interpretation:

Discuss potential overfitting given the high performance in external validation.

4.Improve the abstract’s readability; ensure it clearly states the clinical impact.

5.Check grammar and syntax throughout, especially in the Introduction and Discussion.
**Reviewers' Comments:**
Reviewer's Responses to Questions

**Comments to the Author**

1. Does this manuscript meet PLOS Digital Health’s publication criteria? Is the manuscript technically sound, and do the data support the conclusions? The manuscript must describe methodologically and ethically rigorous research with conclusions that are appropriately drawn based on the data presented.

Reviewer #1: Yes

Reviewer #2: Yes

Reviewer #3: Yes

2. Has the statistical analysis been performed appropriately and rigorously?

Reviewer #1: Yes

Reviewer #2: Yes

Reviewer #3: Yes

3. Have the authors made all data underlying the findings in their manuscript fully available (please refer to the Data Availability Statement at the start of the manuscript PDF file)?

Reviewer #1: Yes

Reviewer #2: No

Reviewer #3: Yes

4. Is the manuscript presented in an intelligible fashion and written in standard English?

Reviewer #1: Yes

Reviewer #2: Yes

Reviewer #3: Yes

5. Review Comments to the Author

Reviewer #1: This study presents an innovative approach integrating 2D and 3D intratumoral heterogeneity scores derived from CT imaging with clinicoradiologic features to predict invasiveness in clinical stage I lung adenocarcinoma. The use of a multicenter cohort, independent external validation, and interpretable machine learning methods are notable strengths. The results demonstrate promising predictive performance. However, several methodological details need clarification, and the biological interpretation of the heterogeneity scores warrants further discussion. With major revisions, the manuscript could be considered for publication.

1. The introduction should more clearly articulate the novelty of the 3D ITH score compared to existing 2D methods and habitat imaging. A direct comparison with prior studies (e.g., Li et al., 2023) would help contextualize the contribution.

2. The use of different CT scanners and protocols across centers may introduce variability. Were any image harmonization techniques (e.g., ComBat, normalization) applied before feature extraction? If not, please discuss as a limitation.

3.The manuscript should provide a clear specification of the optimization criterion used for hyperparameter tuning. It must explicitly state whether optimal hyperparameters were selected based on the Area Under the Curve (AUC) or Accuracy (ACC) metric obtained from out-of-fold (OOF) validation. The rationale behind the choice of this primary evaluation metric, and its role in guiding the hyperparameter search process, should be justified and elaborated.

4.The methodological description of the SHAP-based ablation experiment requires greater clarity. A stepwise sequence should be explicitly outlined: starting with the computation of absolute SHAP values for features, followed by the formulation of an optimization objective based on these values (e.g., sequential backward elimination), and concluding with a detailed, procedural account of how the ablation was carried out in practice.

5.Equation 3 should be reformulated to explicitly indicate that the optimization process is initiated after computing the absolute SHAP values for all candidate models. The revised equation, along with accompanying text, must clearly delineate its applicability across all evaluated models and articulate the logical flow from SHAP value computation to the subsequent feature subset optimization.

6.To bolster the translational relevance of the proposed model, the Discussion should expand on its potential clinical utility. Key points for elaboration include: the technical and practical feasibility of embedding the model into standard radiological reporting workflows; a comparative analysis of its cost-effectiveness versus conventional diagnostic pathways; and a discussion of anticipated barriers to clinical deployment (e.g., integration with hospital IT systems, user acceptance) alongside proposed mitigation strategies.

7.The Discussion should delve deeper into the factors contributing to the superior performance of the CatBoost classifier. This analysis could encompass CatBoost's inherent algorithmic strengths (such as ordered boosting, efficient handling of categorical features, and mechanisms to curb overfitting), its specific suitability for the characteristics of the study dataset (considering feature dimensionality, sample size, and data distributions), and a clear comparison of its performance benefits (e.g., superior AUC/ACC, enhanced cross-validation stability) relative to the other tested models.

8.To improve model interpretability, the inclusion of a SHAP beeswarm plot is recommended. This visualization should serve two primary purposes: firstly, to rank features by their importance based on mean absolute SHAP values; and secondly, to depict how the magnitude and direction (positive or negative) of individual feature values influence the model's predictions. These insights are crucial for verifying the clinical plausibility of the main predictors and for reinforcing the model's interpretability and trustworthiness.

9. The study lacks histopathological correlation of the ITH subregions. This should be highlighted as a key future direction to validate the biological relevance of the imaging-based habitats.

Reviewer #2: The 3D ITH employed in this study is a novel approach for quantifying tumor heterogeneity and has been validated on a large-scale multi-center dataset. I recommend that the manuscript be accepted after major revision.

Reviewer #3: Overall, this study addresses a clinically important problem in the preoperative assessment of lung adenocarcinoma invasiveness and presents a carefully conducted multicenter analysis. The authors propose a three-dimensional intratumoral heterogeneity (3D ITH) quantification framework, extending conventional two-dimensional, single-slice–based approaches to a volumetric representation of tumor spatial complexity. This conceptual shift from planar to three-dimensional characterization is biologically intuitive and represents a meaningful methodological advance in radiomics-based tumor phenotyping. The study further benefits from the use of multicenter data and an independent external validation cohort, which enhances the credibility and potential generalizability of the findings.

From a methodological perspective, the work demonstrates notable strengths. By explicitly modeling intratumoral heterogeneity in three-dimensional space, the proposed 3D ITH metric captures aspects of tumor architecture that are not accessible through traditional radiomic features derived from a single axial slice. In addition, the authors adopt a hybrid feature optimization strategy that balances predictive performance and feature importance, rather than relying solely on conventional regularization techniques such as LASSO. This strategy contributes to both model performance and interpretability and reflects a thoughtful integration of machine learning methodology with clinical objectives.

Nevertheless, several limitations warrant consideration.First, as the central innovation of the study, the robustness and methodological justification of the 3D ITH metric require further clarification.The computation of 3D ITH relies on voxel-wise radiomic features, unsupervised clustering, and predefined spatial connectivity, all of which may be sensitive to segmentation variability and parameter selection. Notably, while the 2D ITH score is derived using six clusters, the 3D ITH framework partitions the tumor volume into five subregions, and the rationale for employing a smaller number of clusters in a higher-dimensional and spatially more complex setting is not sufficiently discussed. Given that cluster number directly influences subregion granularity and the resulting heterogeneity score, additional justification or sensitivity analysis would strengthen confidence that the observed performance gains are attributable to the three-dimensional spatial modeling rather than to specific parameter choices.

Second, an interesting yet unexpected observation is that the model achieved slightly better performance metrics (e.g., AUC) in the external validation cohort than in the internal test set. While such a phenomenon is not implausible in multicenter studies, it is relatively uncommon and merits further clarification. The absence of confidence intervals, statistical comparisons between validation cohorts, or a discussion of potential contributing factors (such as cohort composition or sample size effects) limits the interpretability of this finding.

Finally, certain aspects of figure presentation could be improved to enhance clarity. For example, in Figure 7, the bar chart and the accompanying rose diagram convey information with unequal levels of explicitness. The lack of clear axes, scales, or detailed explanations for the rose diagram makes it difficult for readers to fully interpret the depicted feature importance and its relationship to the quantitative results shown elsewhere.

In summary, this study presents a well-executed and conceptually innovative approach to characterizing tumor heterogeneity and predicting lung adenocarcinoma invasiveness. Addressing the robustness of the proposed 3D ITH metric, clarifying the performance differences between validation cohorts, and refining the presentation of key figures would further enhance the rigor, transparency, and overall impact of the work.

6. PLOS authors have the option to publish the peer review history of their article (what does this mean?). If published, this will include your full peer review and any attached files.

**Do you want your identity to be public for this peer review?** For information about this choice, including consent withdrawal, please see our Privacy Policy.

Reviewer #1: No

Reviewer #2: No

Reviewer #3: No

**Figure resubmission:**  While revising your submission, we strongly recommend that you use PLOS’s NAAS tool (https://ngplosjournals.pagemajik.ai/artanalysis) to test your figure files. NAAS can convert your figure files to the TIFF file type and meet basic requirements (such as print size, resolution), or provide you with a report on issues that do not meet our requirements and that NAAS cannot fix.

After uploading your figures to PLOS’s NAAS tool - https://ngplosjournals.pagemajik.ai/artanalysis, NAAS will process the files provided and display the results in the "Uploaded Files" section of the page as the processing is complete. If the uploaded figures meet our requirements (or NAAS is able to fix the files to meet our requirements), the figure will be marked as "fixed" above. If NAAS is unable to fix the files, a red "failed" label will appear above. When NAAS has confirmed that the figure files meet our requirements, please download the file via the download option, and include these NAAS processed figure files when submitting your revised manuscript. **Reproducibility:** To enhance the reproducibility of your results, we recommend that authors of applicable studies deposit laboratory protocols in protocols.io, where a protocol can be assigned its own identifier (DOI) such that it can be cited independently in the future. Additionally, PLOS ONE offers an option to publish peer-reviewed clinical study protocols. Read more information on sharing protocols at https://plos.org/protocols?utm_medium=editorial-email&utm_source=authorletters&utm_campaign=protocols

---

## [Decision Letter · Decision Letter 1]

29 Jan 2026

Topologically Distinct 2D and 3D Intratumoral Heterogeneity Scores for Preoperatively Predicting Invasiveness in Stage I Lung  Adenocarcinoma: A Multicenter Study

PDIG-D-25-00799R1

Dear Dr. Zuo,

We are pleased to inform you that your manuscript 'Topologically Distinct 2D and 3D Intratumoral Heterogeneity Scores for Preoperatively Predicting Invasiveness in Stage I Lung  Adenocarcinoma: A Multicenter Study' has been provisionally accepted for publication in PLOS Digital Health.

Best regards,

Guochao Zhang

Guest Editor

PLOS Digital Health

**Additional Editor Comments (if provided):**

Dear Dr. Zuo,

We have carefully reviewed the revised version of your manuscript and the responses to the reviewers' comments. Accordingly, I am pleased to inform you that your manuscript is now conditionally accepted.

To proceed to final acceptance, please perform a final check of your results and ensure the manuscript is fully formatted according to the journal’s guidelines. You may then upload the final version via our submission system.

Thank you for your careful and responsive revisions. We look forward to receiving your finalized manuscript soon.

Best regards

**Reviewer Comments (if any, and for reference):**

Reviewer's Responses to Questions

**Comments to the Author**

1. If the authors have adequately addressed your comments raised in a previous round of review and you feel that this manuscript is now acceptable for publication, you may indicate that here to bypass the “Comments to the Author” section, enter your conflict of interest statement in the “Confidential to Editor” section, and submit your "Accept" recommendation.

Reviewer #1: All comments have been addressed

Reviewer #2: All comments have been addressed

Reviewer #3: All comments have been addressed

2. Does this manuscript meet PLOS Digital Health’s publication criteria? Is the manuscript technically sound, and do the data support the conclusions? The manuscript must describe methodologically and ethically rigorous research with conclusions that are appropriately drawn based on the data presented.

Reviewer #1: Yes

Reviewer #2: Yes

Reviewer #3: Yes

3. Has the statistical analysis been performed appropriately and rigorously?

Reviewer #1: Yes

Reviewer #2: Yes

Reviewer #3: Yes

4. Have the authors made all data underlying the findings in their manuscript fully available (please refer to the Data Availability Statement at the start of the manuscript PDF file)?

Reviewer #1: Yes

Reviewer #2: Yes

Reviewer #3: Yes

5. Is the manuscript presented in an intelligible fashion and written in standard English?

Reviewer #1: Yes

Reviewer #2: Yes

Reviewer #3: Yes

6. Review Comments to the Author

Reviewer #1: Dear Authors,

Thank you for your thorough revision in response to the major-review comments. I have examined the revised manuscript point-by-point and confirm that all concerns have been satisfactorily addressed. The paper’s quality has been markedly improved and now meets the journal’s standards. I therefore explicitly recommend: acceptance for publication. I look forward to seeing your work make a positive impact on the field.

With appreciation,

Referee

Reviewer #2: This study effectively enhances the preoperative prediction of invasiveness in lung adenocarcinoma by introducing a topologically distinct 2D and 3D intratumoral heterogeneity score. Based on multicenter, large-sample data, both internal testing and external validation have demonstrated good results. The study has some valuable aspects in methodology and clinical application, and has reasonably revised most of the issues raised by reviewers; it is recommended for acceptance.

Reviewer #3: The authors have thoroughly addressed all previous comments and substantially improved the clarity, methodological transparency, and presentation quality of the manuscript. The revisions strengthen both the scientific rigor and the interpretability of the proposed framework. All major concerns have been adequately resolved, and the manuscript now meets the standards for publication. I have no further comments and recommend acceptance.

7. PLOS authors have the option to publish the peer review history of their article (what does this mean?). If published, this will include your full peer review and any attached files.

**Do you want your identity to be public for this peer review?** For information about this choice, including consent withdrawal, please see our Privacy Policy.

Reviewer #1: No

Reviewer #2: No

Reviewer #3: No
